# Late Holocene slowdown of the Indian Ocean Walker circulation

Mahyar Mohtadi[1], Matthias Prange[1], Enno Schefuß[1] & Tim C. Jennerjahn [2]

Changes in tropical zonal atmospheric (Walker) circulation induce shifts in rainfall patterns along with devastating floods and severe droughts that dramatically impact the lives of millions of people. Historical records and observations of the Walker circulation over the 20th century disagree on the sign of change and therefore, longer climate records are necessary to better project tropical circulation changes in response to global warming. Here we examine proxies for thermocline depth and rainfall in the eastern tropical Indian Ocean during the globally colder Last Glacial Maximum (19–23 thousand years ago) and for the past 3000 years. We show that increased thermocline depth and rainfall indicate a stronger-than-today Walker circulation during the Last Glacial Maximum, which is supported by an ensemble of climate simulations. Our findings underscore the sensitivity of tropical circulation to temperature change and provide evidence for a further weakening of the Walker circulation in response to greenhouse warming.

[1] MARUM-Center for Marine Environmental Sciences, University of Bremen, 28359 Bremen, Germany. [2] Leibniz Centre for Tropical Marine Research (ZMT) GmbH, 28359 Bremen, Germany. Correspondence and requests for materials should be addressed to M.M. (email: mmohtadi@marum.de)

The Walker circulations are zonal atmospheric overturning cells over the tropical oceans with long-term mean surface westerlies along the equatorial Indian and easterlies over the equatorial Pacific Ocean. Their changes are closely tied to the monsoon systems, El Niño-Southern Oscillation (ENSO) and the Indian Ocean Dipole Mode (IOD)[1, 2]. Thus far, studies on the recent development and future projection of the Walker circulations remain contradictory and suggest either a reduced[2-5] or an enhanced[6-9] Walker circulation in response to global warming. This controversy mainly arises from the shortness of the instrumental data covering only a few decades, and necessitates records of Walker circulation changes from the geological past to better understand the sensitivity of the Walker circulation and the hydrological cycle to temperature change[2-4].

One of the most prominent periods for this purpose is the globally cooler climate of the Last Glacial Maximum (LGM). A multi-model ensemble of climate simulations suggests a drier-than-today LGM climate of the Indo-Pacific Warm Pool (IPWP)[10], where the ascending branches of the Indo-Pacific Walker cells reside. A relatively dry IPWP during the LGM has been attributed to a slowdown of the Walker circulation[11-13]. These studies further indicate that similar to the present-day situation[14, 15], records of rainfall and thermocline depth in the eastern tropical Indian Ocean are the most sensitive diagnostic tools to reconstruct past Indian Walker circulation changes. Presently, the most prominent changes in Walker circulation over the Indian Ocean occur during the IOD years, when the circulation weakens (positive IOD events) or strengthens (negative IOD events). In the eastern tropical Indian Ocean, the thermocline shoals and cools while rainfall decreases during the positive IOD events, and vice versa during the negative IOD events[14-16]. In contrast, modern observations show that temperature and rainfall in Africa and the western Indian Ocean beneath the poorly-defined descending branch of the Indian Walker cell[1] do not respond consistently to changes in the Walker circulation[17, 18]. Thus, reconstructing circulation and rainfall in the eastern tropical Indian Ocean is a critical task to evaluate the model performance in simulating the LGM and future changes in the tropical hydrologic cycle.

In this study, we show that the thermocline was deeper and the amount of rainfall was higher in the eastern tropical Indian Ocean during the LGM compared to the late Holocene. In striking agreement with two climate model simulations, our results suggest that the Walker circulation over the Indian Ocean was stronger during the cooler LGM climate. We infer a further weakening of the Walker circulation with increasing global temperatures during the 21st century.

## Results

**Thermocline reconstruction.** Here we present three sea surface temperature (SST) and five thermocline temperature records calculated from shell Mg/Ca of planktic foraminifera at several sites in the eastern tropical Indian Ocean since the LGM (Fig. 1a, Methods and Supplementary Table 1). We make use of the difference between surface and thermocline temperatures ($\Delta T$) to assess the relative depth of the thermocline, with a larger difference indicating a shallower thermocline and vice versa (Fig. 1a, Supplementary Figs. 1–2). The depth of the thermocline in this region is presently controlled by changes in the atmospheric circulation and a sensitive measure of changes in the Walker circulation[14] (Supplementary Figs. 1–2). Similar to the model comparison below, in the following we compare reconstructed average values for the LGM with those of the late Holocene in order to assess changes in the relative strength of the Indian Walker circulation.

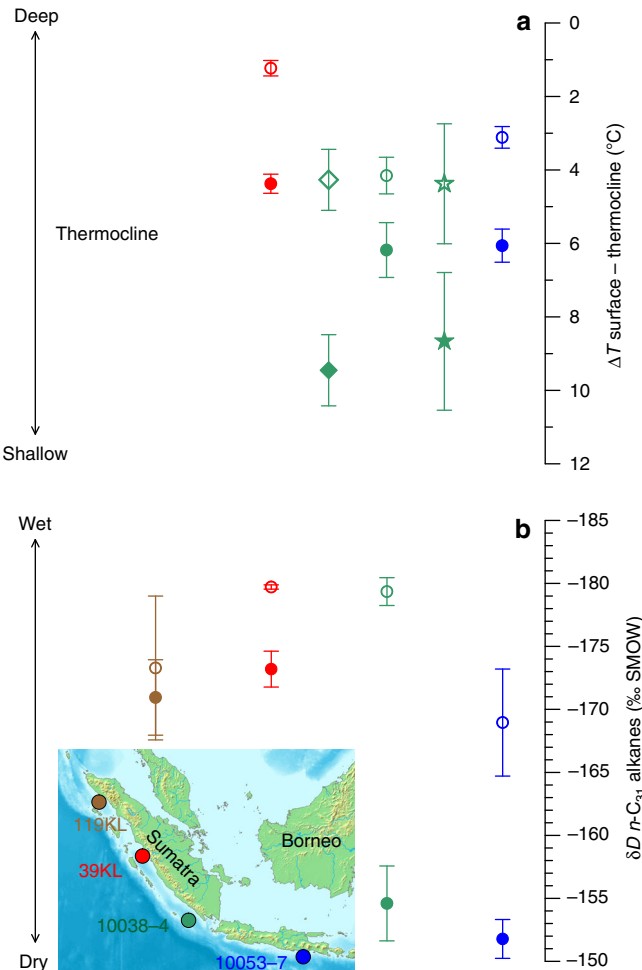

**Fig. 1** Depth of the thermocline and amount of rainfall in the eastern tropical Indian Ocean during the LGM and the late Holocene. **a** Average values of the difference between sea surface and thermocline temperatures ($\Delta T$) as a measure of thermocline depth (reverse scale) at three sites. *Coloured symbols* represent the average values for the late Holocene (*filled*) and the LGM (*open*), and correspond to the sites indicated in the inset map. *Circles* represent the difference between *G. ruber* and *P. obliquiloculata*, *stars* and *diamonds* between *G. ruber* and *N. dutertrei* and *G. tumida*, respectively. *Vertical bars* indicate 95% confidence intervals based on $t$-distributed samples in each time interval (Methods and Supplementary Data 1). The chronology of each period is established independently by [14]C accelerator mass spectrometry dating, each average value comprises a minimum of six samples (Methods, Supplementary Data 1). **b** As in **a** but for the average $\delta D$ values from plant waxes ($n$-$C_{31}$-alkanes) by including an additional record (SO189-119KL). LGM values are corrected for relative changes in ice volume (Methods)

Average temperatures for the late Holocene are about 1.5°–3 °C higher at surface but up to 4 °C lower at the thermocline compared to the average LGM temperatures (Supplementary Data 1). The late Holocene $\Delta T$ values show a coherent picture regardless of the selected temperature calibration or the time span considered (Fig. 2), and reflect the modern conditions with lower values indicating a relatively deeper thermocline in the non-upwelling region off western Sumatra (site 39KL, Fig. 1a, Supplementary Fig. 2).

**Rainfall reconstruction.** In order to reconstruct the amount of rainfall over the eastern tropical Indian Ocean, which is another characteristic feature of past changes in the Indian Walker

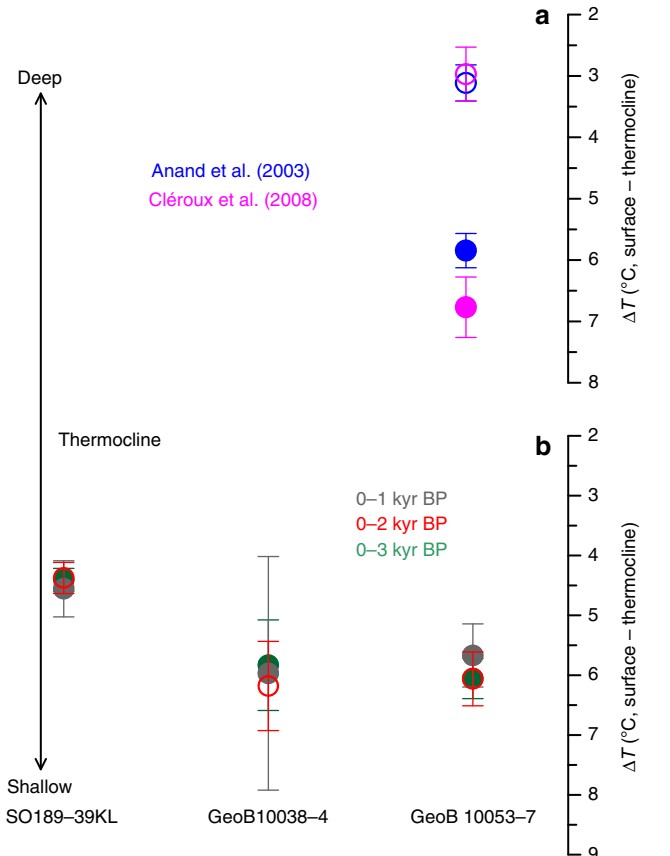

**Fig. 2** Comparison between different temperature calibrations and late Holocene intervals. **a** Difference between average surface and thermocline temperatures ($\Delta T$) for the late Holocene (*filled circles*) and the LGM (*open circles*) using two different temperature calibrations for *P. obliquiloculata*[42, 48] indicated by different colours. *Vertical bars* represent 95% confidence interval based on *t*-distributed samples in each time interval. **b** Average late Holocene difference between surface temperatures derived from Mg/Ca values in *G. ruber* and thermocline temperatures derived from Mg/Ca values in *P. obliquiloculata*. *Grey circles* (*filled*) represent $\Delta T$ for the past 1000 years, *red circles* (*open*) for the past 2000 years and *green circles* (*filled*) for the past 3000 years. *Vertical bars* indicate 95% confidence interval based on *t*-distributed samples in each time interval. Note that the values remain similar regardless of the considered period

circulation[13, 14], we analysed the stable hydrogen isotope composition ($\delta D$) of terrestrial plant waxes (Fig. 1b). In the tropics, a lower $\delta D$ of precipitation indicates an increase in the amount of rainfall[19], which is reflected by lower $\delta D$ values of less degraded plant waxes[20, 21] in our records (Fig. 3). Our results for the late Holocene corroborate this inference and depict the observed present-day spatial pattern in the amount of rainfall that is highest over central Sumatra and decreases slightly towards the northwest, and considerably towards the southeast (Fig. 1b). We corrected the LGM $\delta D$ values for global ice volume and exclude any moisture source other than the Indian Ocean[22] for our sites (see 'Discussion').

**Model simulations**. The dominant role of equatorial zonal wind anomalies in setting the depth of the thermocline in the eastern equatorial Indian Ocean on glacial–interglacial time scale is corroborated by climate model results. A set of LGM simulations from the Paleoclimate Modelling Intercomparison Project

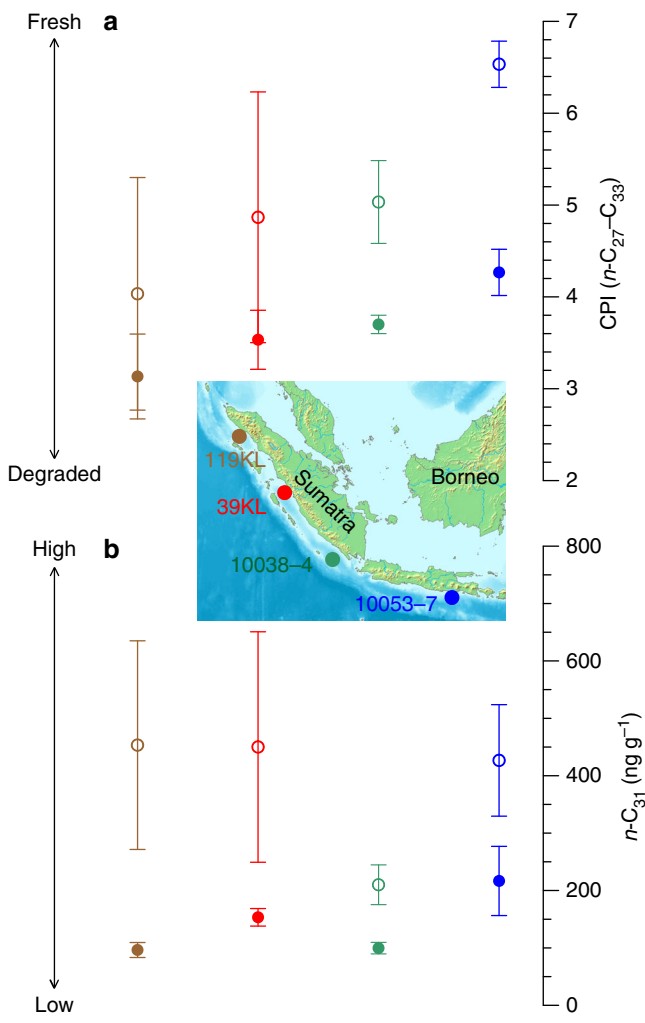

**Fig. 3** Late Holocene and LGM concentration of *n*-alkanes and the CPI values in the eastern tropical Indian Ocean. **a**, **b** Filled and open circles represent average values for the late Holocene and the LGM, respectively. *Vertical bars* indicate 1$\sigma$ standard deviation. Note that at all sites, the LGM is characterized by higher sedimentary concentrations of less degraded plant waxes

(PMIP) phases 2 and 3[10, 23] (https://pmip2.lsce.ipsl.fr/ and https://pmip3.lsce.ipsl.fr/, Supplementary Table 2) suggests an almost one-to-one relationship ($r^2 = 0.81$, $p < 10^{-4}$) between zonal wind anomalies and thermocline depth in the equatorial Indian Ocean (Fig. 4).

## Discussion

Our LGM to late Holocene $\Delta T$ comparison in the eastern tropical Indian Ocean indicates a considerably deeper thermocline during the LGM and a stronger Walker circulation during a globally cooler climate. Notably, average $\Delta T$ values during the LGM are relatively similar at site GeoB 10038-4, regardless of the species used for thermocline temperature reconstruction (*green open symbols* in Fig. 1a). Modern results based on surface sediments show that the temperature signature carried by thermocline dwellers converge when the thermocline is warmer and deeper, and diverge when the thermocline is shallower and cooler[24]. Therefore, the similarity between the $\Delta T$ values during the LGM additionally supports our inference of a deeper thermocline and a stronger Walker circulation.

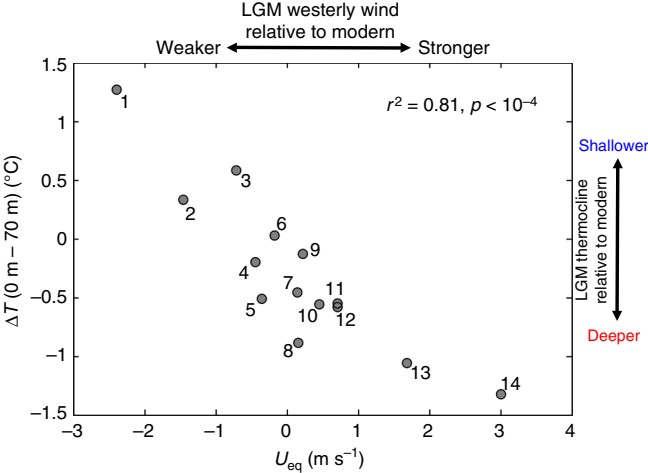

**Fig. 4** LGM thermocline depth anomaly in the eastern Indian Ocean vs. equatorial Indian Ocean zonal wind anomaly as simulated in PMIP2 and PMIP3/CMIP5 models. $\Delta T$ is used as a measure for thermocline depth and defined as the annual mean temperature difference between surface and 70 m averaged over the eastern Indian Ocean region 4° N–6° S, 94° E–104° E. $U_{eq}$ is defined as the annual mean zonal low-level (925 hPa) wind averaged over 5° N–5° S, 50° E–100° E (Fig. 5). Shown are LGM anomalies relative to pre-industrial control runs for both quantities. The models are numbered as in Supplementary Table 2. Models with strongest LGM thermocline deepening and westerly wind anomalies are CCSM3 (No. 13) and FGOALS-g1.0 (No. 14)

The southern part of the study area lies within the Australian-Indonesian monsoon realm with an upwelling system that is governed by the seasonally reversing monsoon winds. In this region, dry southeasterly surface winds originating from the high pressure cell over Australia induce seasonal upwelling during boreal summer from June to September. On average, SST drops from mean annual values of ~ 28 to ~ 25 °C during this season, when the mixed layer is only about 20 m thick (World Ocean Atlas 2009[25], hereafter WOA09). During the rest of the year, the wind direction is generally reversed and moist air is transported east- and southeastward resulting in downwelling and high SST of >28 °C, a relatively thick mixed layer around 70 m and a deep thermocline (WOA09).

The two southern records of $\Delta T$ (sites GeoB10038-4 and 10053-7) lie within the upwelling area of the eastern Indian Ocean and thus, are additionally affected by changes in the meridional Hadley circulation and the southeasterly monsoon winds[26–29]. Reconstructions of marine productivity that is related to the upwelling intensity in this region[30, 31] indicate higher productivity during the late Holocene compared to the LGM[26, 32]. A stronger upwelling and a shallower thermocline in the eastern Indian Ocean during the late Holocene would make the surface cooling more effective and weaken the convective activity and consequently, slowdown the Walker circulation[1, 13]. This scenario can be best observed presently during the IOD events, when anomalous southeasterly winds shoal the thermocline in this area and weaken the Walker circulation over the Indian Ocean[1, 14] (Supplementary Figs. 1–2).

The northern part of the study area off west Sumatra lies outside the Australian-Indonesian monsoon realm and is considered a deep tropical non-upwelling region with year-round high rainfall and a deep thermocline (see also ref. [29]). As part of the Indo-Pacific Warm Pool, mean annual SST in this region is above 28 °C with little seasonal variability. Observation and model studies show that the thermocline depth and rainfall in the

eastern tropical Indian Ocean are the most sensitive diagnostic tools for detecting changes in the zonal atmospheric circulation and convection in the Indian Ocean, i.e. the Indian Walker circulation[13]. We note that the $\Delta T$ results from our site SO189-39KL from a non-upwelling region that is not affected by ocean circulation[29, 33] indicate that any potential scenario must involve a stronger Walker circulation during the LGM compared to the late Holocene.

We consider the Indian Ocean as the primary moisture source for rainfall at our sites[22], both for the late Holocene and the LGM[34]. The subduction of the Indian Ocean Plate beneath the Sunda Plate forms the ~ 3000 km long and up to 3800 m high volcanic arc mountain chains of the Barisan Mountains in Sumatra to the Priangan and the Dieng Mountains in Java since the Oligocene[35]. These orographic barriers inhibit the inflow of low-level winds from moisture sources other than the Indian Ocean to the study area. The blocking effect of these mountains is supposed to change only on tectonic timescales, i.e. millions of years, rendering changes in the moisture source of this region during the LGM unlikely.

Our $\delta D$ results from four sites that lie beneath the ascending branch of the Indian Walker circulation suggest a higher rainfall amount during the LGM compared to the late Holocene (Fig. 1b), and are consistent with another regional $\delta D$ reconstruction[34]. Furthermore, the higher concentrations of less degraded plant waxes in the LGM samples (Fig. 3, Supplementary Data 1) suggest a higher supply of plant debris in the LGM compared to the late Holocene and indicate a higher vegetation cover and/or discharge into the ocean. In combination, these findings indicate higher rainfall over the eastern tropical Indian Ocean and combined with a deeper thermocline, a stronger Walker circulation over the Indian Ocean during the LGM compared to the late Holocene. It is noteworthy that the spatial pattern of change in our $\delta D$ and $\Delta T$ records, with a similar sign of change at all sites but with the highest rate of change off southwest Sumatra resembles hydrological changes related to ENSO and IOD, when substantial changes in the Indian Walker circulation occur (Supplementary Fig. 1).

In the two glacial climate model simulations with maximum westerly surface wind anomalies along the equatorial Indian Ocean (CCSM3 and FGOALS-g1.0) the deepening of the thermocline is sufficiently strong to induce warmer-than-today eastern equatorial Indian Ocean subsurface temperatures in the LGM (Supplementary Fig. 3), consistent with our proxy records, while LGM subsurface cooling in the eastern Indian Ocean is simulated by the other models (not shown). The westerly wind anomalies are associated with a strengthening of the Indian Walker circulation implying anomalous ascent of air over the eastern equatorial Indian Ocean and anomalous subsidence over the western portion (Fig. 5), in accordance with our records from the eastern tropical Indian Ocean (Fig. 1). An anti-correlation of the zonal surface wind anomalies with upper-tropospheric zonal wind anomalies aloft ($r^2 = 0.55$, $p < 10^{-2}$) confirms the involvement of the zonal overturning Indian Walker circulation (Supplementary Fig. 4).

While FGOALS-g1.0 produces strongly enhanced ascent almost everywhere over the Maritime Continent (Fig. 5d), anomalous ascent in CCSM3 is restricted to the western part of the Maritime Continent and the adjacent seas, whereas anomalous subsidence is simulated over the eastern region of the Maritime Continent (Fig. 5b). This scenario agrees with our findings but is at odds with most LGM simulations of the regional tropical circulation and hydroclimate[11–13]. We infer that during the LGM, convection and rainfall over the western part of the IPWP was stronger than today as a result of a stronger Walker circulation, while further to the east, anomalous subsidence

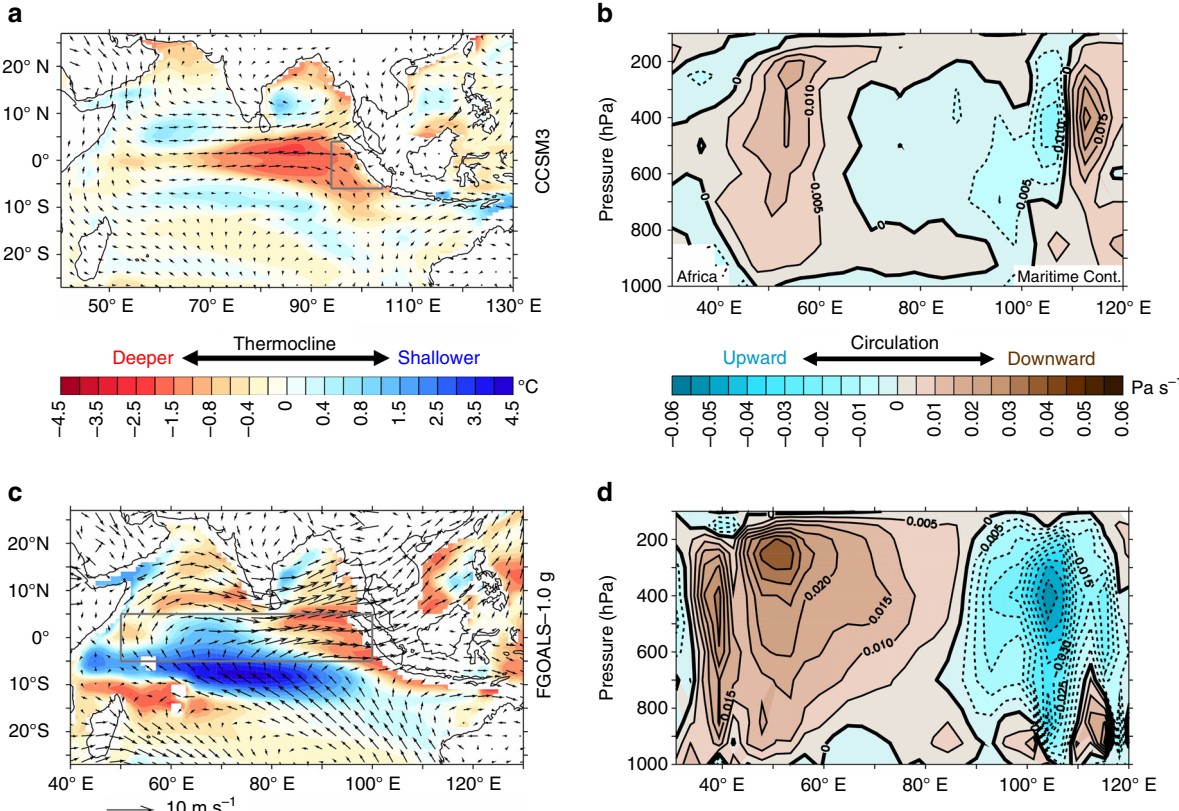

**Fig. 5** LGM climate anomalies in the Indian Ocean region as simulated by CCSM3 and FGOALS-g1.0. Shown are LGM anomalies (annual mean) relative to pre-industrial control runs for $\Delta T$ (defined as ocean temperature difference between surface and 70 m as a measure for tropical thermocline depth) and low-level (925 hPa) wind (**a**, **c**) and vertical velocity ($\omega$) between 5° S and 5° N (**b**, **d**) for the two climate models with greatest LGM thermocline deepening in the eastern equatorial Indian Ocean and strongest equatorial westerly wind anomalies (Nos. 13 and 14 in Fig. 4), CCSM3 (**a**, **b**) and FGOALS-g1.0 (**c**, **d**). Grey box in **a** shows the region for calculating area-averaged $\Delta T$ in all the models in Fig. 4, while the grey box in **c** shows the region for averaging $U_{eq}$. Modern coastlines are drawn in **a** and **c** for better orientation

resulted in drier conditions over the Maritime Continent, as indicated by various proxy and model studies (cf. ref. [13]).

In our model analysis, we show that the depth of the tropical eastern Indian Ocean thermocline is well correlated with the equatorial zonal wind strength and hence a reliable indicator for changes in the Indian Ocean Walker circulation (Fig. 4). In an earlier study[12], the strength of the LGM Indian Ocean Walker circulation was inferred from a compilation of hydroclimate proxies in the Indo-Pacific region and comparison to PMIP model output. There, model agreement with the proxies was quantified by using a Cohen's $\kappa$ statistic, which is defined as the observed fractional agreement relative to the probability of random agreement. Surprisingly, we find that there is no significant correlation ($r^2 = 0.36$, $p > 0.05$) between LGM zonal wind anomalies over the equatorial Indian Ocean and the degree of model-proxy agreement for rainfall reconstructions as quantified by the published maximum Cohen's $\kappa$ (Supplementary Fig. 5). The same holds true for sea surface salinity reconstructions compiled in the same study ($r^2 = 0.25$, $p > 0.05$).

Besides the fact that modelling the Maritime Continent rainfall is generally a challenge for climate models[36], we surmise that three major factors are responsible for the absence of correlation between the synthesized hydroclimate reconstruction and the Indian Ocean Walker circulation: firstly, the proxy data set considered previously[12, 13] comprises a too large region (25° S–20° N, 25° E–170° E) containing many sites that are outside the influence of the Indian Ocean Walker cell; secondly, at many sites contained in the proxy compilation rainfall is controlled by

orographic effects not resolved in the relatively coarse-resolution PMIP climate models, e.g. along the mountainous areas of western Sumatra that receive moisture from the Indian Ocean[22]; hence, regional-scale rainfall patterns are generally simulated with low skill-level in global climate models; lastly, several proxies included in those compilations[12, 13] are not straightforwardly related to local precipitation rate. For instance, surface salinity and rainfall decrease in the eastern tropical Indian Ocean during a weak Walker circulation year, and increase during a strong Walker circulation year, while salinity does not change considerably in the western Indian Ocean in both years[15], thus questioning the feasibility of salinity as a reliable measure of rainfall changes.

Our results from two robust proxies for wind strength and precipitation changes from sites beneath the ascending branch of the Indian Ocean Walker cell are at odds with most of the IPWP climate simulations for the LGM[10, 23]. However, our model-data approach provides a scenario that reconciles the discrepancy in the data and model results of the present and past hydroclimate in this region. Moreover, the inferred scenario of a weaker Walker circulation during the late Holocene from our proxy records, as also simulated by CCSM3 and FGOALS-g1.0, is similar to the projected changes of the Walker circulation over the Indian Ocean during the 21st century greenhouse warming[1, 37]. We note that several forcings besides temperature were different during the LGM compared to the late Holocene, such as ice sheet and insolation, and require further numerical experiments studying the impact of each of these forcings on Walker circulation

changes. Despite different forcing factors, our results provide evidence for the theoretically projected changes in the strength of the tropical circulation during the 21st century[2, 3] and suggest that the zonal circulation will further slow down with continued warming. Finally, a weaker Indian Walker circulation as recorded in our data resembles a positive Indian Ocean Dipole state and supports the projected increase in the frequency of extreme positive IOD events in the 21st century, implying severe impacts on hydroclimate around the Indian Ocean and beyond[37].

## Methods

**Sampling and chronology.** Piston cores SO189-119KL (3° 31′ N, 96° 19′ E; 780 cm core length, 808 m water depth) and SO189-39KL (0° 47′ N, 99° 54′ E; 1350 cm core length, 517 m water depth) were collected from offshore Sumatra during the R/V SONNE cruise 189 in 2006. Gravity cores GeoB 10038-4 (5° 56′ S, 103° 15′ E, 901 cm core length, 1819 m water depth) and GeoB 10053-7 (8° 41′ S, 112° 52′ E, 750 cm core length, 1375 m water depth) were collected from the southern Mentawai Basin offshore southwest Sumatra (10038-4) and off southeast Java (10053-7) during the R/V SONNE cruise 184 in 2005. Piston core SO189-39KL was sampled at 2 cm steps; piston core SO189-119KL and gravity cores GeoB 10038-4 and GeoB 10053-7 at 5 cm steps. Core GeoB 10053-7 was additionally sampled at 2 cm steps between 400 cm and 750 cm core depth corresponding to 22–8 kyr BP[28]. For this study, only the LGM (19–23 kyr) and the late Holocene (0–3 kyr) sections of these cores are considered. Age models of the cores were published previously[27–29]. The LGM (late Holocene) sections contain 2 (2) radiocarbon datings in SO189-119KL[29], 7 (9) in SO189-39KL[29], 1 (1) in GeoB 10038-4[38] and 3 (3) in GeoB 10053-7[28]. Sedimentation rates during the LGM and the late Holocene are similar in SO189-119KL (~ 20 cm kyr$^{-1}$), in SO189-39KL (~ 40 cm kyr$^{-1}$) and in GeoB 10038-4 (~ 10 cm kyr$^{-1}$). Sedimentation rates during the LGM (late Holocene) are around 40 (60) cm kyr$^{-1}$ in GeoB 10053-7.

**Planktic foraminifera and thermocline reconstruction.** All cores used in this study lie above the calcite lysocline and contain well-preserved aragonitic pteropods that suggest a negligible effect of dissolution on planktic foraminifera shell geochemistry. Previous studies on surface sediments[24] and sediment trap time-series[39] from the eastern tropical Indian Ocean suggest that mean calcification depth for the mixed-layer dwelling species *Globigerinoides ruber* is about 20 m. Mean calcification depths of the thermocline species have been estimated in the same studies at about 75 m for *Pulleniatina obliquiloculata*, at 75–95 m for *Neogloboquadrina dutertrei* and at about 100 m for *Globorotalia tumida*. Three out of eight temperature records have been published previously[27–29]. For Mg/Ca analyses, a minimum of 30 *G. ruber* specimens from the 250 to 355 µm size-fraction and a minimum of 20 specimens of the remaining species from the 355 to 500 µm size-fraction have been selected, crushed and cleaned following a slightly modified protocol of Barker et al.[40] with five water and two methanol washes, two oxidation steps with 1% NaOH-buffered $H_2O_2$ and a weak acid leach with 0.001 M quartz distilled (QD) $HNO_3$. Samples were then dissolved into 0.075 M QD $HNO_3$ and centrifuged for 10 min at 6000 r.p.m., transferred into test tubes and diluted. Mg/Ca ratios in samples from core GeoB 10038-4 and GeoB 10053-7 were measured with a Perkin Elmer Optima 3300R Inductively Coupled Plasma Optical Emission Spectrophotometer (ICP-OES) equipped with an auto sampler and an ultrasonic nebulizer U-5000 AT (Cetac Technologies Inc.). Mg/Ca ratios in samples from core SO189-39KL were measured with an Agilent Technologies 700 Series ICP-OES with a CETAX ASX-520 auto sampler. Both facilities are housed at the Faculty of Geosciences, University of Bremen. Mg/Ca values are reported as mmol mol$^{-1}$. Instrumental precision was determined using an external, in-house standard (Mg/Ca = 2.92 mmol mol$^{-1}$) and the ECRM 752-1 standard[41], which were run after every fifth and fiftieth sample, respectively. Relative standard deviations for the external standard, ECRM 752-1 standard, and replicate measurements are listed in Supplementary Table 1. Cleaning efficiency was monitored by measuring Fe/Ca, Mn/Ca and Al/Ca ratios (<0.1 mmol mol$^{-1}$ for Mn/Ca and Fe/Ca, and not detectable for Al/Ca). Mg/Ca ratios were converted to temperature using the following equations:

For *G. ruber*[42]:

$$\text{Mg/Ca} \, (\text{mmol mol}^{-1}) = 0.38 \exp^{(0.09*T \, °C)} \quad (1)$$

For *P. obliquiloculata*[42]:

$$\text{Mg/Ca} \, (\text{mmol mol}^{-1}) = 0.328 \exp^{(0.09*T \, °C)} \quad (2)$$

For *N. dutertrei*[42]:

$$\text{Mg/Ca} \, (\text{mmol mol}^{-1}) = 0.342 \exp^{(0.09*T \, °C)} \quad (3)$$

For *G. tumida*[24]:

$$\text{Mg/Ca} \, (\text{mmol mol}^{-1}) = 0.41 \exp^{(0.068*T \, °C)} \quad (4)$$

The use of other calibrations would change the absolute values but not the pattern of change that is central to this study, as demonstrated in Fig. 2. Core-top studies from hypersaline regions such as the Mediterranean Sea with salinities

above 36 psu suggest that the Mg/Ca ratio in planktic foraminifera shells is additionally controlled by salinity[43]. Data from the IPWP core tops do not show a salinity effect on Mg/Ca as the 36 psu threshold is out of reach in this rain-laden region[44]. Errors (1$\sigma$) were calculated following Gibbons et al.[44] and Mohtadi et al.[29], and the 95% confidence intervals are based on $t$-distributed samples in each time interval and indicated by *vertical bars* (Fig. 1a, Supplementary Data 1).

Results from modern samples corroborate the use of the difference between the Mg/Ca-based temperatures of a surface- and a thermocline-dweller ($\Delta T$) for reconstructing changes in the thermocline depth. This difference in the sediment trap time-series JAM1-2 deployed off south Java[39] is smallest during the northwest monsoon season (~2 °C), when high precipitation is accompanied by a warm and deep thermocline[39]. In contrast, the largest difference (~5 °C) occurs when thermocline is shallow and cool during the dry southeast monsoon season[39]. Likewise, Mg/Ca-based temperature estimates in 70 surface sediment samples from this region show that the $\Delta T$ between *G. ruber* and *P. obliquiloculata* or *N. dutertrei* follows changes in the stratification of the water column and consequently, the thermocline depth[24].

**Plant wax analyses.** We measured three sediment samples per core and period to average internal variability within each time slice (Supplementary Data 1). About 10 g of dried and ground sediments were extracted with a Dionex Accelerated Solvent Extractor (ASE-200) at 100 °C and 1000 psi with a mixture of dichloromethane/methanol (9:1) for 5 min, which was repeated three times. Squalane was added before extraction as internal standard. Asphaltenes were removed from the total lipid extracts (TLEs) by elution with hexane over $Na_2SO_4$. TLEs were saponified by 6% KOH in methanol and acid fractions removed. Neutral fractions were separated into aliphatic (apolar), ketone and polar fractions on a silica gel column. Afterwards the aliphatic fraction was separated on an $AgNO_3$-impregnated silica-column into saturated and unsaturated fractions, of which the first fraction contains the $n$-alkanes. Quantification of $n$-alkanes was performed by gas-chromatography–flame ionization detection (GC–FID) on a ThermoFisher Scientific Focus Gas chromatograph. Alkanes were identified and quantified by comparison of retention times and peak areas with an external standard mixture. Repeated analyses of the external standard mixture yield a quantification uncertainty of <5%. The $n$-$C_{31}$ alkane was the most abundant homologue in all samples. Concentrations varied from 90 to 640 ng g$^{-1}$ dry sed. and were consistently higher in LGM than Holocene samples (ED 5–6). The carbon preference index (CPI) was calculated to:

$$\text{CPI} = 0.5 * \left( \sum C_{\text{odd}27-33} \Big/ \sum C_{\text{even}26-32} + \sum C_{\text{odd}27-33} \Big/ \sum C_{\text{even}28-34} \right) \quad (5)$$

with $C_x$ the amount of each homologue[45].

CPI values of individual samples varied between 2.6 and 6.8 (Supplementary Data 1). On average, the CPI values for the LGM time-slices were higher than for the Holocene samples at each site (Fig. 3). In conjunction, sedimentary concentrations and CPI values indicate a higher contribution of undegraded, i.e. directly plant-derived, long-chain $n$-alkanes to the LGM sediments compared to the Holocene.

Hydrogen isotope compositions were analysed on a ThermoFisher Scientific Trace gas chromatograph connected to a ThermoFisher Scientific MAT 253 mass spectrometer via a pyrolysis reactor operated at 1420 °C. Isotope values were measured against calibrated $H_2$ reference gas. δD values are reported in ‰ relative to VSMOW. The $H^{3+}$ factor was determined daily and varied around 6.23 ± 0.04 ppm nA$^{-1}$ during analyses. An alkane standard of 16 externally calibrated alkanes was measured after every 6th measurement. Long-term precision and accuracy of the external alkane standard were 3 and <1‰, respectively. All samples were run at least in duplicate. δD values of the $n$-$C_{31}$ alkane varied from −150 to −174 ‰ (Supplementary Data 1). Reproducibility for the $n$-$C_{31}$ alkane was on average 1‰ (<1 to 4‰). Precision and accuracy of the squalane internal standard were 6 and 4‰, respectively ($n = 34$). Mean values and average standard deviation of δD values were calculated for each time slice and are reported within 95% confidence intervals. We correct δD values for global seawater isotopic changes during the LGM. The ice volume correction[46] was converted to δD values using the global meteoric water line[47]. The ice-volume correction shifts the δD values during the LGM by −9.1‰.

**Model simulations.** To test the relation between glacial anomalies in zonal equatorial winds and thermocline depth in the Indian Ocean we use LGM and pre-industrial coupled climate model simulations from PMIP2 and PMIP3/CMIP5[10, 23]. LGM boundary conditions include reduced greenhouse gases, changed astronomical parameters, continental ice sheets and a changed land–sea mask in accordance with a ~ 120 m lower global sea level compared to today (see https://pmip2.lsce.ipsl.fr and https://pmip3.lsce.ipsl.fr for details). See Supplementary Table 2 for climate models used in PMIP2 and PMIP3/CMIP5. All model analyses are based on climatological annual means.

**Data availability**. The authors declare that the data supporting the findings of this study are available within the paper and its supplementary information files. Data can also be downloaded at https://doi.pangaea.de/10.1594/PANGAEA.877994

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

## Acknowledgements

We are grateful to A. Lückge, B. Meyer-Schack, M. Segl, H. Kuhnert, R. De Pol-Holz, R. Kreutz, S. Pape, K. Gesierich, M. Baumer and B. Beckmann for laboratory assistance. This study was supported by the Bundesministerium für Bildung und Forschung through grants 03G0184A (PABESIA), 03G0189A (SUMATRA), 03G0806B (CARIMA) and the PalMod initiative, and the Deutsche Forschungsgemeinschaft (grants JE281/4-1, HE3412/15-1, and the DFG Research Centre/Cluster of Excellence 'The Ocean in the Earth System'). We acknowledge the World Climate Research Programme's Working Group on Coupled Modelling, which is responsible for CMIP, and the climate modelling groups (listed in Supplementary Table 2) for producing and making available their model output. For CMIP the U.S. Department of Energy's Program for Climate Model Diagnosis and Intercomparison provides coordinating support and led development of software infrastructure in partnership with the Global Organization for Earth System Science Portals. The Laboratoire des Sciences du Climat et de l'Environnement (LSCE) is acknowledged for collecting and archiving the PMIP2 model data. The PMIP2 Data Archive is supported by CEA, CNRS and the Programme National d'Etude de la Dynamique du Climat (PNEDC).

## Author contributions

M.M. and E.S. generated and analysed the data, M.P. analysed the climate model experiments. M.M. and T.C.J. designed the study and all authors discussed and interpreted the data.

**Additional information**

**Competing interests:** The authors declare no competing financial interests.

