## [Peer Review File · Nature Communications]

Reviewers' comments:

Reviewer #1 (Remarks to the Author):

Mohtadi et al. use a suite of proxy records (SST, thermocline temp, and leaf wax) to compare thermocline depth and rainfall between the LGM (a cooler climate) and late Holocene to gain insight into the response of the eastern Indian Ocean portion of the Walker Circulation (WC) to global temperature change. They provide a convincing argument that this type of comparison/study is useful and needed, considering that climate models predict a slowdown of WC, but studies based on observational data are inconsistent and contradictory. Mohtadi et al. find evidence of a deeper thermocline and greater rainfall, which suggests a stronger Walker Cell during LGM than late Holocene. They use this to suggest that we can expect the Walker Cell to continue to weaken under further warming in the 21st century.

Overall, this paper is well-written and the results, particularly from the proxy records seem consistent with one another, and mostly significant. I am left with a few questions that are relatively minor – mainly concerning some assumptions made, information on figures, and some suggestions for areas where I think the paper would benefit from further explanation/clarification. Once these issues are addressed, I believe this study is relevant and suitable for publication.

Main Manuscript/Figure Questions:

1. Fig 1a - It looks like at site 10038-4 that the three (green/open) LGM thermoclines, despite differing species, are very consistent with one another. Can you comment on why the late Holocene (green/filled) data varies from one another? Obviously it's clear that the takeaway from this figure is that LGM thermoclines were relatively (and significantly) deeper than those of late Holocene, however I wonder why species did not matter for LGM but did matter for late Holocene?
2. Fig 1b – Any hunch as to why there would be little (no significant) change between LGM and late Holocene at the most northern site (119KL)? Does the WC usually affect precipitation at this location and if so, what are some explanations for why a shift in WC between the two time periods would not show up in this record? The authors' discussion on lines 107-110 is helpful to understanding the overall pattern, but I'm not sure it entirely answers this particular question regarding record 119KL.
3. Fig 3 – The arrows are a little difficult to see without zooming in a lot. Additionally, why are the boxes used for averaging ΔT and U_{eq} different between the two models?
4. The authors introduce all PMIP model runs, but then spend the rest of the paper focusing on the two with maximum westerly wind anomalies, presumably because those two are consistent with the proxy data analyzed earlier in the study. However (unless I missed it), this point isn't explicitly made until lines 143-145 of the paper near the concluding remarks. I think it would be helpful for readers if the authors were more explicit earlier on in the paper about why the focus shifts to CCSM3 and FGOALS-g1.0.
5. I have a question about the sentence on lines 147-150. The authors state, "Moreover, our results provide evidence for the theoretically projected changes in the strength of the tropical circulation during the 21st century and suggest that the zonal circulation will further slow down with continued warming." My question is: How appropriate is it to extrapolate your conclusions about late Holocene to projected changes in the 21st century. This conclusion rests heavily on the assumption that the warming that occurred between LGM and late Holocene (0-3 ka BP) affected the dynamics in the same way that the dynamics are affected currently by greenhouse gas forcing. I'm not arguing that it's inappropriate, but I think it's a large enough assumption that it warrants more discussion within the paper, especially since the authors explicitly reference the 21st century.
6. I'm left wanting a bit more from the conclusions in general. The last sentence of the article, which states that a weaker Walker Circulation is consistent with a positive IOD state is important and could be elaborated upon and/or brought up earlier in the paper.

Extended Data/Figure Questions:

1. ED2 – Where is GeoB 10028-2 and SO189-MS01? How is SO189-MS01 used for two different sites (SO189-39KL and GeoB 100038-4)?
2. ED4 & ED6 – “dots” and “circles” may confuse some readers. I suggest using the terms “filled” and “open” as the authors did in Figure 1.

Reviewer #2 (Remarks to the Author):

Review of « Response to the Indian Ocean Walker Circulation to global temperature change » by Mohtadi et al (Nature Communications)

This manuscript presents an original analysis of paleoceanographic and paleoclimatic data, as well as paleoclimate simulations over the equatorial eastern Indian Ocean. The thermocline depth is inferred by reconstructing temperatures for the surface and at ~70m depth via the use of surface dwelling and thermocline dwelling foraminifera. These reconstructions are complemented by reconstructions of precipitation on the adjacent island of Sumatra, based on deuterium measured on plant waxes. The reconstructions suggest a deeper thermocline and enhanced precipitation at LGM compared to the late Holocene, compatible with a stronger Walker Circulation and at odds with the results from a previous compilation (DiNezio and Tierney, 2013). This analysis on paleodata is complemented with model results, using the Paleoclimate Modelling Intercomparison Project (PMIP) simulations. The model data is key to show the physical link between thermocline depth, surface winds, and large scale atmospheric circulation. It also shows that the relationship between precipitation and surface wind is not straightforward, which could explain why the results of the present study differ from those of DiNezio and Tierney, which were partly based on salinity reconstructions.

This work is original because it takes advantage of the different living depths of key foraminifera to reconstruct the vertical structure of the ocean temperature near the surface, presents the marine and terrestrial reconstructions in a physically consistent framework via the analysis on model results. However, some clarifications are needed before the manuscript can be published:

- the authors present *Pulleniatina obliquiloculata*, *Neogloboquadrina dutertrei*, *Globorotalia tumida* as thermocline dwellers and reconstruct temperatures at their typical present habitat depth. It is not clear to me that these species would have stayed at these depths at LGM, if the thermocline has deepened. This needs to be clarified for the results to be fully convincing. Also, the relationship between $\Delta T(\text{surface} - 70\text{m})$ and the thermocline depth should be better proven, maybe through present data. The basic idea is promising but its demonstration should be improved.

- the authors allude to changes in productivity (lines 77ff) and directly relate these changes in productivity to changes in upwelling strength and thermocline depth. But could other factors explain the changes in productivity? temperature? light? nutrient supply. This needs to be better explained or removed.

- the rainfall reconstruction based on plant wax δD is used to advocate for stronger convection, and a stronger Walker cell. But model results show that rainfall changes do not correlate very well with circulation changes and the authors use this argument to infer that the salinity reconstructions used by previous studies to document hydroclimatic changes over these regions are not valid. This inconsistency should be resolved.

- one strength of the previous reconstructions was to consider reconstructions on a larger area, at the scale of the Walker cell itself. Since the authors claim to be relating their local results to the large scale, it would be relevant to discuss their results w.r.t results from a larger zone. The other option would be to keep the focus on the eastern equatorial Indian Ocean, which would be perfectly fine by me, as these results are already very new, original and promising to build

constrains on models from paleodata on this very region.

In brief, I find these results promising and worth a publication in Nature Communications, but work is needed to further clarify the points above to make the story clearer and fully consistent.

Minor comments:

- The authors should acknowledge the PMIP and CMIP projects (<https://pmip2.lsce.ipsl.fr/> and <http://cmip-pcmdi.llnl.gov/cmip5/citation.html>).

- Extended data 4, caption: "dots" and "circles" should be replaced by "open" and "filled" circles, as there are the symbols used on the figure

- Figure 1: the maps shows the location of the cores. it could also show basic climate features for the story, such as thermocline depth and precipitation

- Figures 2 and 3: why considering a box which extends so far eastward? Is the result robust for slightly different regions. It would be good to justify how this region was defined. Also, having an idea of the interannual variability on Fig 2 would help showing how different the model results are. This comment is also valid for extended data 9 and 10.

Reviewer #1:

Main Manuscript/Figure Questions:

1. Fig 1a - It looks like at site 10038-4 that the three (green/open) LGM thermoclines, despite differing species, are very consistent with one another. Can you comment on why the late Holocene (green/filled) data varies from one another? Obviously it's clear that the takeaway from this figure is that LGM thermoclines were relatively (and significantly) deeper than those of late Holocene, however I wonder why species did not matter for LGM but did matter for late Holocene?

Response:

This is a very well-taken point and we refrained from explaining this in the original version due to space limitation. What we find in the average late Holocene values in this core (GeoB 10038-4, green filled symbols in Fig. 1a) is in agreement with results from two studies on modern archives in this area, on surface sediments (Mohtadi et al., 2011, ref. #24) and sediment traps (Mohtadi et al., 2009, ref. #39). The thermocline at site GeoB 10038-4 is presently relatively shallow and steep (see Supplementary Fig. 2b). Surface sediment records in the southern Mentawai Basin suggest an average temperature difference of 2 to >4°C between these species (see the left panel in the below figure, from Mohtadi et al., 2011). This is the same range as in Fig. 1a for the late Holocene (filled symbols).

The situation during the LGM, when all three values converge (open symbols in Fig. 1a), is comparable to present-day conditions at site SO189-39KL with a deep but gentle thermocline (Supplementary Fig. 2a). Surface sediment records in the Northern Mentawai Basin suggest an average temperature difference of only 0 to <3°C between these species (see the right panel in the below figure, from Mohtadi et al., 2011). Simplified, modern data suggest the deeper the thermocline, the smaller the difference between thermocline and surface, but also among the thermocline species. This finding additionally supports our interpretation of a stronger Walker circulation during the LGM, when thermocline was deeper at site GeoB10038-4 than during the late Holocene.

We add this information in the revised version, lines 101-108 and 297-307.

2. Fig 1b – Any hunch as to why there would be little (no significant) change between LGM and late Holocene at the most northern site (119KL)? Does the WC usually affect precipitation at this location and if so, what are some explanations for why a shift in WC between the two time periods would not show up in this record? The authors' discussion on lines 107-110 is helpful to understanding the overall pattern, but I'm not sure it entirely answers this particular question regarding record 119KL.

Response:

We agree with the referee that the change at the northern site 119KL is little and not significant. Unfortunately, we do not have a thermocline temperature record from this site to confirm (or reject) whether this is also the case for the upper ocean conditions. As mentioned in our manuscript in line 152, we find the same sign of change in another site published by Niedermeyer et al. (2014, ref. #34) that lies between our two northern sites (119KL and 39KL). Therefore we are confident in the sign of change, although the magnitude of change is less than in the other, southerly sites. On the other hand, this is in accordance with present-day observation of Walker circulation changes, e.g. during the positive Indian Ocean Dipole events, with larger changes south of the equator (see lines 159-162).

3. Fig 3 – The arrows are a little difficult to see without zooming in a lot. Additionally, why are the boxes used for averaging ΔT and U_{eq} different between the two models?

Response:

We have opted for the same vector length scale in both figures 5a and 5c (formerly Fig. 3). This shows that the wind anomalies are generally larger in FGOALS (Fig. 5c) than in CCSM3 (Fig. 5a), which is important to see. If we increased the arrow lengths, figure 5c would become impossible to decipher, hence we are limited by the outcome of these two models with respect to wind anomalies. With the present scaling, we think that the wind anomalies along the equator (which is the focus of our study) are best seen in both figures.

As for the size of the boxes, they are not different between the models. We show one box (in Fig. 5a) to show the averaging area for temperature, while the other box (in Fig. 5c) shows the averaging area for wind. Plotting both boxes into one figure would make them hard to see. We rephrased the figure caption for clarification (penultimate sentence in the figure caption).

4. The authors introduce all PMIP model runs, but then spend the rest of the paper focusing on the two with maximum westerly wind anomalies, presumably because those two are consistent with the proxy data analyzed earlier in the study. However (unless I missed it), this point isn't explicitly made until lines 143-145 of the paper near the concluding remarks. I think it would be helpful for readers if the authors were more explicit earlier on in the paper about why the focus shifts to CCSM3 and FGOALS-g1.0.

Response:

We agree and add this information whenever possible, in lines 49, 168, and 172-173.

5. I have a question about the sentence on lines 147-150. The authors state, "Moreover, our results provide evidence for the theoretically projected changes in the

strength of the tropical circulation during the 21st century and suggest that the zonal circulation will further slow down with continued warming.” My question is: How appropriate is it to extrapolate your conclusions about late Holocene to projected changes in the 21st century. This conclusion rests heavily on the assumption that the warming that occurred between LGM and late Holocene (0-3 ka BP) affected the dynamics in the same way that the dynamics are affected currently by greenhouse gas forcing. I’m not arguing that it’s inappropriate, but I think it’s a large enough assumption that it warrants more discussion within the paper, especially since the authors explicitly reference the 21st century.

Response:

We agree with the referee that the cause of the 21st century warming is different than between the LGM and the late Holocene, with the former being primarily forced by anthropogenic greenhouse gas emission. However, since we observe the same features in both scenarios, which involve a shallower thermocline and less rainfall in the eastern tropical Indian Ocean during the warmer late Holocene (in our records) and during the 21st century warming (instrumental records and model projections, see ref. #37), we would expect the same consequences such as increased positive IOD events and a further weakening of the Walker cell over the Indian Ocean. Likewise, the model simulations also show anomalies that are similar to those projected for the 21st century and suggest the same dynamics, although the forcing might be different. We add this discussion in the revised version (lines 159-162, 220-224).

6. I’m left wanting a bit more from the conclusions in general. The last sentence of the article, which states that a weaker Walker Circulation is consistent with a positive IOD state is important and could be elaborated upon and/or brought up earlier in the paper.

Response:

We did as suggested by the referee and elaborate this briefly in the revised version, lines 36-40, 159-162 and 220-224.

Extended Data/Figure Questions:

1. ED2 – Where is GeoB 10028-2 and SO189-MS01? How is SO189-MS01 used for two different sites (SO189-39KL and GeoB 100038-4)?

Response:

These CTD casts were deployed at the same sites as SO189-39KL and GeoB 100038-4, respectively. This is indicated in each of the two panels by “At site...” (now Supplementary Fig. 2).

We do not use SO189-MS01 for two different sites. For SO189-39KL, the corresponding CTD cast is SO189-MS40, deployed on September 15, 2006 (blue text in panel a).

2. ED4 & ED6 – “dots” and “circles” may confuse some readers. I suggest using the terms “filled” and “open” as the authors did in Figure 1.

Response:

Changed in the revised version as suggested by the referee (now Figs. 2 and 3).

Please note that in order to comply with format requirements of *Nature Communications*, almost the entire supplementary text is now integrated in the main text.

Reviewer #2:

- the authors present *Pulleniatina obliquiloculata*, *Neogloboquadrina dutertrei*, *Globorotalia tumida* as thermocline dwellers and reconstruct temperatures at their typical present habitat depth. It is not clear to me that these species would have stayed at these depths at LGM, if the thermocline has deepened. This needs to be clarified for the results to be fully convincing. Also, the relationship between ΔT (surface – 70m) and the thermocline depth should be better proven, maybe through present data. The basic idea is promising but its demonstration should be improved.

Response:

The first point has also been raised by the first referee, though phrased differently. Based on surface sediment results the temperature signature carried by thermocline dwellers converge when thermocline is warmer and deeper, and diverge when thermocline is shallower and cooler (see also our response to the first comment by referee #1). This observation-based relationship between the temperature signal and the thermocline depth is most probably related to changing habitat depth as a function of nutrient availability being different at the two settings, as correctly addressed by the referee. So probably the habitat depth of these species also converged during the LGM, at least by judging from their temperature signal. However, a habitat depth change must involve changes in the water column structure, and as indicated by modern data, the temperature signals carried by these species converge when thermocline is deeper. Hence, the fact that the LGM values of these three species are almost identical further supports our inference that the Walker circulation was stronger and the thermocline was deeper during the LGM. We add this information in the revised version, lines 101-108.

As for the second point, we think that the relationship between ΔT and thermocline depth is best illustrated in ED 2 (now Supplementary Fig. 2). While ΔT at sites 10038-4 (39KL) decreases by 2°C (3.5°C) during a strong Walker circulation year (red in Supplementary Fig. 2), the thermocline deepens by 30 m (40 m). We also discuss the modern data in lines 297-307. These findings are also shown in ARGO profiles published by Qiu et al. (2012, ref. #15, see Fig. in response to referee's 3rd comment). There, the isothermal and mixed layers (black and white lines) deepen during the strong Walker circulation scenario (year 2010, climatological means are represented by dashed lines), while the deeper thermocline warms by up to 4°C east of 80°E (gray lines).

- the authors allude to changes in productivity (lines 77ff) and directly relate these changes in productivity to changes in upwelling strength and thermocline depth. But could other factors explain the changes in productivity? temperature? light? nutrient supply. This needs to be better explained or removed.

Response:

Changes in productivity, temperature, or nutrient supply in this region are inherently related to changes in upwelling strength and thermocline depth. This is inferred from a number of physical and chemical oceanography studies from this region (see e.g.

the special issue in *Oceanography*, 2005, volume 18, issue 4, “The Indonesian Seas”). We think that explaining this would be outside the focus of this study and thus, refrain from discussing this in further details and add ref. #30-31 for modern conditions, and refer to ref. #26 and 32, which discuss this relationship for the LGM. We understand that one option would be to remove this part completely, as suggested by the referee. However, our argument is that even if changing upwelling intensity is the cause of changes in the thermocline depth (as suggested by paleoproductivity studies), the effect would be the same, i.e. a more effective surface cooling and decreased convective activity during the late Holocene, and ultimately a weaker Walker circulation. We note that due to reformatting of this manuscript to comply with the journal’s requirements, this section is much longer than in the original version (now lines 109-140)

- the rainfall reconstruction based on plant wax δD is used to advocate for stronger convection, and a stronger Walker cell. But model results show that rainfall changes do not correlate very well with circulation changes and the authors use this argument to infer that the salinity reconstructions used by previous studies to document hydroclimatic changes over these regions are not valid. This inconsistency should be resolved.

Response:

Modern data show that the relationship between Walker circulation, rainfall, and salinity is not straightforward. This is why we think that salinity is not a good indicator of changes in the Walker circulation. According to results based on ARGO profiles from a strong year (2010) and a weak year (2006) of Walker circulation, Qiu et al. (2012, ref. #15) state for the strong Walker circulation scenario: “The westerly wind anomalies drive a stronger Wyrтки Jet... carrying more salty water eastward. Thus, despite a negative E-P..., there is a dramatic increase of salinity in the upper layer in the east”. For the weak Walker circulation scenario, they state: “a weaker Wyrтки Jet in response to the same equatorial easterly anomalies... reduces the eastward transport of salty water, lowering salinity in the upper layer (upper ~50 m) east to 65° E”. These salinity anomalies are shown below by colors (Fig. From Qiu et al., 2012). We make this clear in the revised version, in lines 211-215.

- one strength of the previous reconstructions was to consider reconstructions on a larger area, at the scale of the Walker cell itself. Since the authors claim to be relating their local results to the large scale, it would be relevant to discuss their results w.r.t results from a larger zone. The other option would be to keep the focus on the eastern equatorial Indian Ocean, which would be perfectly fine by me, as these results are already very new, original and promising to build constraints on models from paleodata on this very region.

Response:

We completely agree with the referee with respect to the two possible options. We opt for the second option (only data from the eastern equatorial Indian Ocean), as modern results suggest that in the western equatorial Indian Ocean the impact of changing Walker circulation is not consistent everywhere (lines 40-43, ref. #1, 15, 17, 18), let alone in other areas that are additionally affected by other climate phenomena. We indicate this in lines 203-205, 210-215.

Minor comments:

- The authors should acknowledge the PMIP and CMIP projects (<https://pmip2.lscce.ipsl.fr/> and <http://cmip-pcmdi.llnl.gov/cmip5/citation.html>).

Response:

Included in the revised version as suggested (lines 94 and 359-360)

- Extended data 4, caption: “dots” and “circles” should be replaced by “open” and “filled” circles, as there are the symbols used on the figure

Response:

Changed in the revised version as suggested (now Fig. 2).

- Figure 1: the maps shows the location of the cores. it could also show basic climate features for the story, such as thermocline depth and precipitation

Response:

In theory, this is a valuable suggestion but hardly workable. Firstly, the inset map in Fig. 1 is too small for this purpose. We think that the inset map nicely shows the topographic barrier with respect to the moisture source, as discussed in lines 141-149. Secondly, we could not find any appropriate dataset containing “thermocline depth” over a larger area. The available datasets (20°S isotherm, the mixed-layer depth, or the isothermal depth) are too coarse and not representative of thermocline depth (e.g. the average thermocline temperature in this area is much higher than 20°C, and due to the existence of a barrier layer, the depth of the mixed- or isothermal layer is not equivalent to the thermocline depth). Lastly, from our δD records we infer the sign of change in rainfall, not its absolute value. We mention, show, or refer to basic climate features on numerous occasions, e.g. in Supplementary Figs. 1 and 2, lines 113-114, 117-118, 133-134).

- Figures 2 and 3: why considering a box which extends so far eastward? Is the result robust for slightly different regions. It would be good to justify how this region was defined. Also, having an idea of the interannual variability on Fig 2 would help showing how different the model results are. This comment is also valid for extended data 9 and 10.

Response:

It is not fully clear to us which box the reviewer is referring to. The box for averaging ocean temperatures mainly includes the relevant proxy sites along the coast of Sumatra. The box for wind-averaging mainly refers to the lower limb of the Indian Ocean Walker circulation. Note that these boxes are used for all models but shown separately for the sake of clarity. When doing the analyses we played a while with the sizes of the boxes, but did not find a sensitive dependence of the wind-thermocline correlation (shown in Fig. 4) to the exact sizes of the two boxes. If the size of the maps in Fig. 5 is referred to, we think that extending the longitudinal range farther east is important to show the vertical velocity pattern over the Maritime Continent, which is helpful to explain the discrepancy among the proxy records from this area. We indicate this in lines 177-181 and 184-186.

We agree with the reviewer that it would be nice to calculate interannual variability for the variables shown in Fig. 2, ED 9 and ED 10 (now Fig. 4 and Supplementary Figs. 4 and 5). Unfortunately, this is not possible with the files provided in the PMIP databases. These files only include climatological monthly means for a certain field (e.g. zonal velocity u) along with variance at each grid point (again for each month, i.e. 12 values for every grid cell). So we do not have long time-series from which interannual variability could be calculated for an area-averaged quantity. It is needless to say that the area-averaged variance of a variable (e.g. zonal wind u) is NOT the variance of the area-averaged variable. However, all climate fields provided in the PMIP databases are based on long-term means, such that we are very confident that most differences shown in these figures are statistically significant.

Please note that in order to comply with format requirements of *Nature Communications*, almost the entire supplementary text is now integrated in the main text.

REVIEWERS' COMMENTS:

Reviewer #1 (Remarks to the Author):

I am extremely pleased with the level of detail and clarify with which the authors addressed both mine and the other referee's issues. I have no further edits or suggestions for the reviewers and am confident this will be a valuable contribution and recommend it for publication.

Reviewer #2 (Remarks to the Author):

Review of Mohtadi et al., second version

The second version of the manuscript is much improved compared to the initial one and makes the conclusions stronger. The constructive comparison to the previous synthesis from DiNezio is very useful and helps establishing a more detailed view of the atmospheric circulation in this area for the LGM.

I have a few last comments which can be addressed quickly before publication:

The presentation of the general context of this work pre-supposes that the reconstructed changes in the Walker cell are due to the LGM temperatures being cooler than present. It is true that it is a strong forcing, but other changes cannot, from the PMIP experiments, be excluded. For instance the fact that the Sahul and Sunda shelves are exposed (such as assumed by DiNezio et al) can probably by itself affect the Walker cell. Teleconnections from far away changes in forcings, such as ice sheets, cannot be excluded either. It would be good that the authors include a few sentences of discussion stating that further numerical experiments studying the impact of each of these forcings would be very beneficial to confirm the mechanisms of Walker cell changes at the LGM. These have been recently attempted by, for instance, Klockmann et al, CP, 2016, for other contexts.

Line 86: insert "global" in front of "ice volume"

Could the authors acknowledge the PMIP modelling groups for their efforts? This is much appreciated by the modelling groups and the groups building the infrastructure for all MIPs to happen! The PMIP2 standard acknowledgement can be found here: <https://pmip2.lsce.ipsl.fr/database/access/acknowledge.shtml> and reads: "We acknowledge the international modeling groups for providing their data for analysis, the Laboratoire des Sciences du Climat et de l'Environnement (LSCE) for collecting and archiving the model data, ...add your customized acknowledgements here... The PMIP 2 Data Archive is supported by CEA, CNRS and the Programme National d'Etude de la Dynamique du Climat (PNEDC). The analyses were performed using version mm-dd-yyyy of the database. More information is available on <http://pmip2.lsce.ipsl.fr/>."

The PMIP3-CMIP5 acknowledgements can be found here: <http://cmip.llnl.gov/cmip5/citation.html> and read "We acknowledge the World Climate Research Programme's Working Group on Coupled Modelling, which is responsible for CMIP, and we thank the climate modeling groups (listed in Table XX of this paper) for producing and making available their model output. For CMIP the U.S. Department of Energy's Program for Climate Model Diagnosis and Intercomparison provides coordinating support and led development of software infrastructure in partnership with the Global Organization for Earth System Science Portals."

Reviewer #1:

I am extremely pleased with the level of detail and clarity with which the authors addressed both mine and the other referee's issues. I have no further edits or suggestions for the reviewers and am confident this will be a valuable contribution and recommend it for publication.

Response:

We are grateful to the referee for the insightful comments and suggestions that greatly improved this paper.

Reviewer #2:

The second version of the manuscript is much improved compared to the initial one and makes the conclusions stronger. The constructive comparison to the previous synthesis from DiNezio is very useful and helps establishing a more detailed view of the atmospheric circulation in this area for the LGM.

I have a few last comments which can be addressed quickly before publication:

The presentation of the general context of this work pre-supposes that the reconstructed changes in the Walker cell are due to the LGM temperatures being cooler than present. It is true that it is a strong forcing, but other changes cannot, from the PMIP experiments, be excluded. For instance the fact that the Sahul and Sunda shelves are exposed (such as assumed by DiNezio et al) can probably by itself affect the Walker cell. Teleconnections from far away changes in forcings, such as ice sheets, cannot be excluded either. It would be good that the authors include a few sentences of discussion stating that further numerical experiments studying the impact of each of these forcings would be very beneficial to confirm the mechanisms of Walker cell changes at the LGM. These have been recently attempted by, for instance, Klockmann et al, CP, 2016, for other contexts.

Response:

We agree with the referee and add this information to the discussion (lines 226-229)

Line 86: insert "global" in front of "ice volume"

Response:

Changed as suggested (line 87)

Could the authors acknowledge the PMIP modelling groups for their efforts? This is much appreciated by the modelling groups and the groups building the infrastructure for all MIPs to happen! The PMIP2 standard acknowledgement can be found here: <https://pmip2.lsce.ipsl.fr/database/access/acknowledge.shtml> and reads: "We acknowledge the international modeling groups for providing their data for analysis, the Laboratoire des Sciences du Climat et de l'Environnement (LSCE) for collecting and archiving the model data, ...add your customized acknowledgements here... The PMIP 2 Data Archive is supported by CEA, CNRS and the Programme National d'Etude de la Dynamique du Climat (PNEDC). The analyses were performed using version mm-dd-yyyy of the database. More information is available on <http://pmip2.lsce.ipsl.fr/>.

The PMIP3-CMIP5 acknowledgements can be found here: <http://cmip.llnl.gov/cmip5/citation.html> and read “We acknowledge the World Climate Research Programme's Working Group on Coupled Modelling, which is responsible for CMIP, and we thank the climate modeling groups (listed in Table XX of this paper) for producing and making available their model output. For CMIP the U.S. Department of Energy's Program for Climate Model Diagnosis and Intercomparison provides coordinating support and led development of software infrastructure in partnership with the Global Organization for Earth System Science Portals.”

Response:

We included this in Acknowledgements, and thank the referee for the thoughtful comments and suggestions that considerably improved this paper.